# Revisiting the Role of LXRs in PUFA Metabolism and Phospholipid Homeostasis

**DOI:** 10.3390/ijms20153787

**Published:** 2019-08-02

**Authors:** Antoine Jalil, Thibaut Bourgeois, Louise Ménégaut, Laurent Lagrost, Charles Thomas, David Masson

**Affiliations:** 1Université Bourgogne Franche-Comté, LNC UMR1231, F-21000 Dijon, France; 2INSERM, LNC UMR 1231, F-21000 Dijon, France; 3FCS Bourgogne-Franche Comté, LipSTIC LabEx, F-21000 Dijon, France

**Keywords:** liver X receptors, phospholipids, polyunsaturated fatty acids, lipogenesis, inflammation

## Abstract

Liver X receptors (LXRs) play a pivotal role in fatty acid (FA) metabolism. So far, the lipogenic consequences of in vivo LXR activation, as characterized by a major hepatic steatosis, has constituted a limitation to the clinical development of pharmacological LXR agonists. However, recent studies provided a different perspective. Beyond the quantitative accumulation of FA, it appears that LXRs induce qualitative changes in the FA profile and in the distribution of FAs among cellular lipid species. Thus, LXRs activate the production of polyunsaturated fatty acids (PUFAs) and their distribution into phospholipids via the control of FA desaturases, FA elongases, lysophosphatidylcholine acyltransferase (LPCAT3), and phospholipid transfer protein (PLTP). Therefore, LXRs control, in a dynamic manner, the PUFA composition and the physicochemical properties of cell membranes as well as the release of PUFA-derived lipid mediators. Recent studies suggest that modulation of PUFA and phospholipid metabolism by LXRs are involved in the control of lipogenesis and lipoprotein secretion by the liver. In myeloid cells, the interplay between LXR and PUFA metabolism affects the inflammatory response. Revisiting the complex role of LXRs in FA metabolism may open new opportunities for the development of LXR modulators in the field of cardiometabolic diseases.

## 1. Introduction

Beside their prominent role in controlling cholesterol homeostasis, regulation of fatty acid (FA) metabolism appears to be an important function of liver X receptors (LXRs), and pivotal studies have highlighted major lipogenic activity of LXRs [1,2,3]. All the enzymes involved in the biosynthesis of saturated or monounsaturated fatty acids are regulated by LXRs. Indeed, acetyl-CoA carboxylase (ACC), fatty acid synthase (FASN), and steroyl-CoA desaturase (SCD1) have been identified as direct LXR targets [4,5,6]. LXRs may also regulate lipogenesis indirectly by controlling the expression of key transcription factors such as carbohydrate-response element-binding protein (ChREBP), sterol responsive element binding protein 1c (SREBP1c), and peroxisome proliferator activated receptor gamma (PPAR gamma) [2,7,8]. Consequently, in vivo stimulation of LXR activity by synthetic agonists such as T0901317 induces a dramatic activation of FA and triglyceride synthesis in the liver in different mouse models resulting in hepatic steatosis, stimulation of VLDL secretion, and increased plasma triglyceride levels [1,9,10,11]. This hepatic steatosis, also observed in humans during a recent clinical trial, is a major undesirable side effect, which has, to date, limited the therapeutic development of synthetic LXR agonists [12]. The lipogenic activity of LXRs has also been characterized in different cells/tissues including subcutaneous adipose tissue, pancreas, skeletal muscle cells, and macrophages [13,14,15,16,17,18]. LXRs mainly induce the synthesis of monounsaturated fatty acid, notably oleic acid (C18:1 n-9) and palmitoleic acid (C16:1 n-7) due to the coordinated induction of FASN and SCDs [11]. These FAs are subsequently incorporated in triglycerides and phospholipids. Overall, it results in an increased monounsaturated-to-saturated-FA ratio in cells and a modulation of cell membrane properties [11,16]. Interestingly, very recent studies have suggested that besides this well-characterized role in monounsaturated FA synthesis, LXRs promote the synthesis of long-chain n-3 and n-6 polyunsaturated fatty acids (PUFAs). Moreover, LXRs also regulate key enzymes involved in phospholipid metabolism including lysophosphatidylcholine acyltransferase 3 (LPCAT3) and the plasma phospholipid transfer protein (PLTP) [19,20,21,22]. Thus, by controlling the synthesis of biologically active PUFAs as well as their distribution into cellular glycerophospholipids, LXRs could be able to modulate the biological properties of cell membranes and the release of PUFA-derived mediators. In accordance with this hypothesis, recent works have identified several biological functions of LXRs that are directly mediated by modulation of PUFA and phospholipid metabolism.

## 2. LXR and PUFA Synthesis

The synthesis of long-chain PUFAs is a complex process, involving the initial activation of the substrate FA into an acyl-CoA. Subsequently, several steps of elongation and desaturation occur that result in the formation of very-long-chain PUFAs of the n-3 or n-6 families such as arachidonic acid (AA), eicosapentaenoic acid (EPA), or docosahexaenoic acid (DHA) (Figure 1). These PUFAs are incorporated into cellular lipids, notably glycerophospholipids that are particularly enriched with n-3 and n-6 PUFAs. Recently, it was demonstrated that the activation of LXRs by synthetic agonists induced, in a coordinated manner, the expression of all the enzymes required for PUFA synthesis, including activation (ACSL3), delta 6 and delta 5 desaturation (FADS2 and FADS1), and FA elongation (ELOVL5) [17]. Interestingly, most of these genes are co-regulated by LXRs and SREBP1c and it appears that LXR and SREBP1c co-bind at specific sites to control the expression of PUFA genes depending on the metabolic context (Figure 2) [23,24].

### 2.1. Fatty Acid Activation

As aforementioned, the activation of the FA substrate into acyl-CoA via a thioesterification reaction is an important limiting step for PUFA synthesis. This reaction is catalyzed by a class of enzyme called acyl-CoA synthases. Although there are at least five long-chain fatty acyl CoA synthases (ACSLs), they display differential tissue distribution and specificity toward FAs. Two members of the family, ACSL3 and ACSL4, are regulated by LXRs. ACSL3 exhibits a preference for C18-C20 PUFAs over saturated and monounsaturated fatty acids [25]. The regulation of ACSL3 by LXR has been observed in placental trophoblast cells, macrophages, and the liver [17,26,27]. ACSL3 is directly regulated by LXR via a functional LXRE that has been described in the gene promoter. An indirect regulation of ACSL3 by LXR through SREBP1c also occurs [17,24]. The activation of ACSL3 by LXR induces an overall increase of acyl-CoA synthase activity and uptake of FAs by the cells [27]. ACSL4, an arachidonate-preferring acyl-CoA synthase, is also induced by synthetic LXR ligands and desmosterol in mouse macrophages while the LXR-regulation in human cells appears less pronounced [23,28].

### 2.2. Elongation

The limiting step in FA elongation is the condensation of a malonylCoA to an acylCoA molecule, which is performed by elongation of very-long-chain FAs proteins (ELOVLs). ELOVLs 2, 4, and 5 are preferentially associated with PUFA metabolism. Several studies have identified ELOVL5 as an LXR target in macrophages and hepatocytes while ELOVL2, more specifically involved in DHA synthesis does not seem to be regulated by LXRs [17,29,30]. ELOVL5 displays a preference for C18 to C20 n-3 and n-6 PUFAs, leading to the synthesis of C20-C22 PUFAs such as arachidonic acid (AA) (C20:4 n-6), adrenic acid (C22:4 n-6), and docosapentaenoic acid (DPA) (C22:5 n-3). ELOVL5 is regulated directly by LXR but also indirectly through the LXR-SREBP1c axis [17,24,29]. At basal state, in murine macrophages, the *Elovl5* gene is constitutively repressed by nuclear corepressor (Ncor) binding to LXR on the *Elovl5* gene promoter. Accordingly, Ncor deletion results in a derepression of *Elovl5* gene that subsequently promotes the synthesis of long-chain n-3 PUFAs with anti-inflammatory properties [30].

### 2.3. Desaturation

The coordinated action of elongases with desaturases orchestrates the synthesis of very-long-chain PUFAs. The desaturation reactions occur specifically at positions 5 and 6 of the fatty acyl chain and are catalyzed by delta 5 and delta 6 desaturases (FADS1 and FADS2, respectively). Strikingly, no other gene in the human or murine genomes is able to compensate for their activity. *FADS1* and *FADS2* are well-recognized SREBP1c target genes and their expression is strongly down regulated by long-chain PUFAs through the inhibition of SREBP1c [31,32]. FADS1 and FADS2 are also regulated by LXRs either directly or indirectly through SREBP1c. Stimulation of macrophages with synthetic LXR ligands such as GW3965 or T0901317 results in an induction of *FADS1* and *FADS2* gene expression [17]. In contrast, a steroidal agonist such as desmosterol inhibits FADS1 and FADS2 likely due to the ability of desmosterol to suppress SREBP1 processing [17]. Unlike *ELOVL5*, *FADS1* and *FADS2* genes appear to be strongly dependent on SREBP1c for basal expression levels but *FADS1* and *FADS2* are also regulated by LXR binding on their gene promoters (Figure 2). As observed for ELOVL5, a basal repression of *FADS1* and *FADS2* by NCOR binding to LXR on their promoter is present [30].

## 3. LXRs in Phospholipid Transfer and Remodeling

Cellular PUFAs are mainly present as esterified forms. Neutral lipids such as esterified cholesterol and triacylglycerol contain significant amounts of PUFAs but phospholipids (PLs) represent the most dynamic pool within the cells. The PUFA content of PLs directly contributes to the physicochemical properties of biological membranes. Moreover, the release of PUFAs from PLs during inflammatory processes leads to the synthesis of pro- and anti-inflammatory mediators. Thus, the role of LXRs in regulating PL metabolism appears to be complementary to their action on PUFA biosynthesis.

### 3.1. LPCAT3

PLs are continuously remodeled through deacylation and reacylation reactions. This process is called the Lands cycle [33]. In this metabolic pathway, the turnover of PUFAs at the *sn*-2 position is mediated by the reciprocal actions of phospholipases A2 and lyso-PL acyltransferases [34]. One member of the family, i.e., LPCAT3, is the major LPCAT isoform in macrophages, in the intestine, and the liver; LPCAT3 exhibits a strong preference for C20 PUFAs such as AA and EPA over C18 and C22 PUFAs [22]. LPCAT3 seems to be a key determinant of cell membrane homeostasis in liver and in intestine [35,36]. In macrophages, LPCAT3 contributes to the regulation of AA distribution and eicosanoid release upon LXR stimulation. *LPCAT3* is a direct LXR target gene, and a functional LXRE has indeed been described close to the transcription start site (Figure 2) [19,22]. Activation of LXR by synthetic agonist increases LPCAT3 activity in macrophages and liver and results in an enrichment of PLs with PUFAs, including AA [22,37]. In contrast to other genes involved in PUFA metabolism, LPCAT3 does not seem to be regulated by SREBP1c as *SREBP1c* knockdown does not affect LPCAT3 induction following LXR agonist treatment [17]. 

### 3.2. PLTP

The phospholipid transfer protein (PLTP) belongs to lipid transfer/lipopolysaccharide binding protein family (LT-LBP). PLTP is a secreted protein and has been mainly studied for its role in lipoprotein metabolism. [38] PLTP acts as a carrier that transfers phospholipids between lipoprotein particles. PLTP is able to transfer a large range of amphipathic compounds, such as phospholipids, tocopherol (vitamin E), lipopolysaccharides, and free cholesterol [38]. *PLTP* has been identified as an LXR target gene by different groups in the liver and macrophages [20,21] and treatment of mice with synthetic LXR agonist (T0901317, 10 mg/kg) significantly increases plasma PLTP activity. However, SREBP1c seems to be required for LXR-mediated induction of PLTP in mouse liver as this regulation is abolished in *Srepb1c^−/−^* mice [39].

## 4. Control of Phospholipid Metabolism by LXRs in the Liver: Impact on Lipogenesis and VLDL Secretion

As discussed above, a major biological effect resulting from LXR activation by pharmacological agonists is the occurrence of hepatic steatosis associated with the secretion of large VLDLs enriched with triglycerides. Recent studies suggest that PLTP and LPCAT3, two LXR targets involved in phospholipid metabolism, play a critical role in this context. In addition to its function in the intravascular metabolism of lipoproteins, PLTP also plays a role in the intracellular transfer of PLs [40,41]. By transferring PLs to the nascent apo B in the endoplasmic reticulum (ER) and in the Golgi compartments, PLTP contributes to the lipidation of apo B and to VLDL assembly. This function has been extensively documented in the liver, and concordant studies show that inactivation of PLTP results in decreased VLDL secretion, while conversely, liver PLTP overexpression stimulates VLDL secretion [41]. As liver-PLTP expression is under the control of LXRs, it could be hypothesized that PLTP is an effector of LXR in the control of VLDL secretion. Indeed, Okasaki et al. demonstrated that administration of T0901317 no longer increases plasma triglyceride levels in *Ldlr^−/−^*/*Srebp1c^−/−^* mice as compared to *Ldlr^−/−^* mice [39]. Moreover, the induction of PLTP consecutive to LXR agonist treatment (0.0075% T0901317 for six days) was nearly abolished in *Srebp1c^−/−^* mice. Strikingly, PLTP overexpression by using an adenovirus restored partially the ability of T0901317 to induce hypertriglyceridemia and to stimulate the secretion of large-sized VLDL particles in *Ldlr^−/−^* mice lacking *Srebp1c* [39]. 

LPCAT3 is another LXR target involved in phospholipid metabolism. In connection with the ability of LPCAT3 to transfer PUFAs into phospholipids, *Lpcat3*-deficient mice display a decrease in the PUFA content of PLs, notably arachidonic acid, in different tissues, including the liver [35,36]. This decreased PUFA-content is associated with an alteration of the cell membrane properties, including decreases in membrane fluidity and in the ability of membranes to form curve structures [35,36]. At the cellular level, these alterations can induce endoplasmic reticulum (ER) stress during acute LPCAT3 inhibition [37]. More importantly, in the long term, phospholipid remodeling mediated by LPCAT3 appears to be required for appropriate membrane triglyceride transfer and VLDL assembly. Consequently, mice deficient in *Lpcat3* in the liver show reduced plasma triglyceride levels, increased hepatic steatosis, and impaired ability to secrete large size VLDLs. Interestingly, GW3965, a synthetic LXR agonist, very modestly increases VLDL levels in *Lpcat3*-deficient mice as compared to WT mice, indicating that LPCAT3 is required for the induction of VLDL secretion mediated by LXR activation. Importantly, LPCAT3 is not only an effector of LXRs on VLDL secretion but it also appears to control hepatic lipogenesis. Indeed, enrichment of ER with PUFAs controls SREBP1c processing and cleavage [42]. In response to LXR activation, LPCAT3 stimulates the incorporation of PUFAs into phospholipids and subsequently induces a SREBP1c-dependent lipogenic transcriptional program [42]. Accordingly, the lipogenic activity of LXR agonists is markedly reduced in the absence of LPCAT3 expression in the liver.

Both LPCAT3 and PLTP appear to play important roles as effectors of the LXR-SREBP1c axis on hepatic lipogenesis and VLDL secretion. Interestingly, *Pltp* is one of the most induced genes in liver-specific *Lpcat3*^−/−^ mice either on chow or western type diets suggesting a potential compensatory role of PLTP in a context of LPCAT3 deficiency [35]. Future studies will be necessary to investigate the relationship between PLTP and LPCAT3 in the control of liver lipid metabolism. 

Interestingly, a reciprocal regulation of LXR activity by PUFAs with potential consequences on the development of hepatic steatosis has been described. Indeed, several long-chain PUFAs such as AA, EPA, and DHA are potent LXR antagonists and inhibitors of *SREBP1* transcription [43,44]. Moreover, dietary n-3 FA supplementation has been proposed as a potential strategy to limit the hepatic steatosis associated with LXR agonist treatment [45].

## 5. LXRs, PUFA Metabolism and Inflammation

In macrophages, LXRs control the inflammatory response through different mechanisms. It was shown that LXRs suppress the expression of NF-κB and AP-1 responsive genes by a trans-repression mechanism involving a ligand-dependent sumoylation [46]. The anti-inflammatory activity of LXRs is also related to their ability to inhibit TLR/Myd88 signaling through cholesterol efflux and ABCA1-dependent cholesterol depletion of lipid rafts [47]. Finally, LXR suppresses inflammation through direct repression of pro-inflammatory genes and stimulation of cholesterol efflux [48]. It is likely that additional mechanisms, independent of cholesterol efflux, could also be involved in the anti-inflammatory activity of LXR agonists. Indeed, the atheroprotective potential of LXR agonists is preserved in *Ldlr*^−/−^ mice even in the absence of Abca1 and Abcg1 in myeloid cells [49]. While the anti-inflammatory effects of LXRs have been mainly demonstrated in mice, the paradigm in humans is different, and long-term exposure of human macrophages to LXR agonists leads to a potentiation of the LPS response [50].

In this context, the metabolism of PUFA in macrophages seems to be a promising candidate to mediate the effects of LXRs on inflammation. Indeed, PUFA metabolism is central for the control of macrophage functions and the inflammatory response. PUFAs act as precursors of bioactive molecules such as eicosanoids. As components of PLs, they have a pivotal role in cell membrane biology. Finally, PUFAs are also endowed with ligand properties for nuclear or membrane receptors such as GPR120 that regulates inflammation. While the activation of LXRs with synthetic agonists induces the synthesis of n-6 and n-3 PUFAs in macrophages, [17] most of the LXR target genes involved in PUFA metabolism are repressed at a basal state due to a constitutive binding of NCOR to the LXR sites [30]. NCOR deletion in macrophages results in an anti-inflammatory phenotype by inducing, in an LXR-dependent manner, the synthesis of anti-inflammatory n-3 PUFAs. This study, therefore, identified a new molecular mechanism linking LXR and inflammation through n-3 FA metabolism [30].

The same metabolic pathways are used for the synthesis of long-chain n-6 and n-3 PUFAs that possess either pro- or anti-inflammatory properties. Indeed, n-6 FAs and, notably, AA are the precursors of pro-inflammatory mediators such as prostaglandins and leukotrienes. Therefore, by modulating AA synthesis and distribution, LXRs may also affect the release of pro-inflammatory mediators. It was observed that treatment of primary human macrophages with LXR agonist induce both AA synthesis and an enrichment of PLs with AA in an LPCAT3-dependent manner [17,22]. Accordingly, pretreatment of human macrophages with LXR agonist increased the release of arachidonate-derived eicosanoids, such as prostaglandin E2 and thromboxane after LPS stimulation. It suggests that LXR-mediated induction of LPCAT3 controls eicosanoid secretion by increasing the pool of AA, which can be subsequently mobilized from phospholipids [22]. Interestingly, constitutive *Lpcat3* deficiency in mouse macrophages induces a pronounced alteration of AA distribution within the cellular lipids due to impaired incorporation of AA in phospholipids [51]. Long-chain PUFAs such as AA are recognized as LXR antagonists [43] and a repression of LXR pathways, in *Lpcat3^−/−^* macrophages, was observed, including decreased cholesterol efflux. Accordingly, *Lpcat3* deficiency in hematopoietic cells increases atherosclerosis in *Ldlr^−/−^* mice [51].

Finally, an ultimate mechanism linking LXR and inflammation is the modulation of the PUFA composition of phospholipids. It was shown that LXR-LPCAT3 pathway protects against ER stress through regulation of membrane lipid composition in models of acute LPCAT3 inhibition or overexpression [37]. A pro-inflammatory phenotype associated with LPCAT3 deficiency was also observed in link with an increased activity of the c-Src- c-Jun N-terminal kinase (JNK) pathway. The mechanism was related to the recruitment and the subsequent phosphorylation of c-Src-in lipid rafts, secondary to changes in membrane saturation. Similarly, a potentiation of the inflammatory response due to an increase of Tlr4 in lipid rafts and increased phosphorylation of c-Src was observed in macrophages with a partial *Lpcat3* deficiency [52]. However, in another study using the same model, *Lpcat3* deficiency had no effect on *Il1b* or *Cox2* induction following LPS stimulation of murine macrophages treated or not with LXR agonists [48].

## 6. Conclusions

The function of LXR in regulating the metabolism of PUFAs and phospholipids is only emerging. Currently, there is a general context where the role of PUFAs and their derivatives in the development of atherosclerosis and metabolic diseases is reassessed, and omega 3 supplementation appears as a plausible strategy to prevent atherosclerosis [53,54]. Therefore, the ability of LXRs to modulate this metabolism needs to be further investigated. It may open new opportunities for the development of innovative strategies in the field of cardio metabolic diseases.

## Figures and Tables

**Figure 1 ijms-20-03787-f001:**
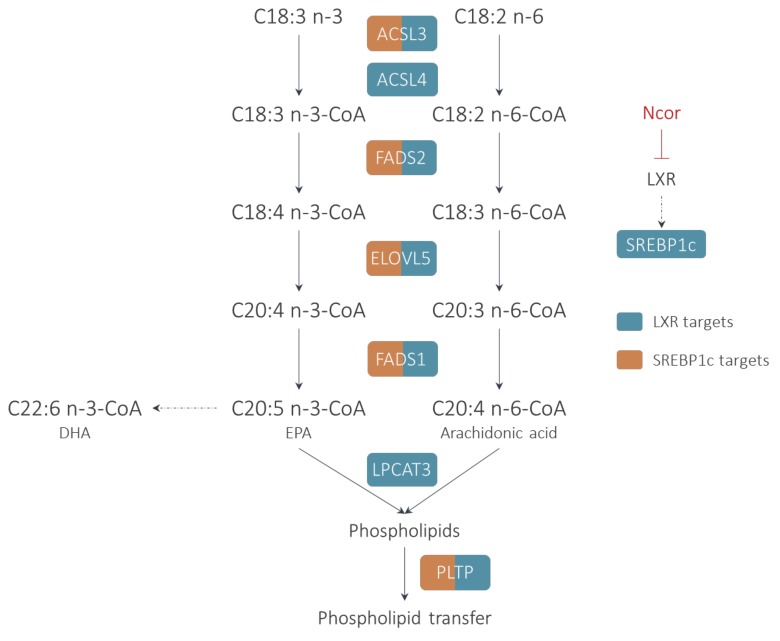
Regulation of polyunsaturated fatty acids (PUFA) synthesis and phospholipid metabolism by liver X receptors (LXRs). PUFA requires the initial activation of the substrate fatty acid into an acyl-CoA by an acyl-CoA synthase (ACSL3 and ACSL4) and the successive actions of delta 5 desaturase (FADS1), elongase (ELOVL5), and delta 6 desaturase (FADS2). LXR regulates directly or indirectly through sterol responsive element binding protein (SREBP) all these classes of enzymes. Interestingly, LXR activation also increases lysophosphatidylcholine acyltransferase (LPCAT3), which promotes the preferential incorporation of eicosapentaenoic acid (EPA) and arachidonic acid (AA) into glycerophospholipids which can be further exchanged by phospholipid transfer protein (PLTP).

**Figure 2 ijms-20-03787-f002:**
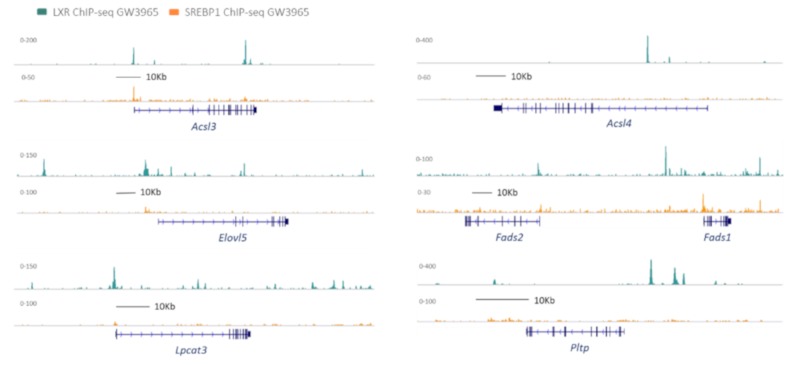
UCSC genome browser tracks depicting LXR and SREBP1 ChIP-Seq peaks (normalized tag counts) at genes involved in PUFA and phospholipid metabolism in mouse macrophages treated with GW3965. Images created from data by Oishi et al. Cell Metab. 2017 Feb 7;25(2):412-427 (GSE79423) [24].

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
