# Peer review of "Revisiting the Role of LXRs in PUFA Metabolism and Phospholipid Homeostasis"

_ijms, 2019, doi:10.3390/ijms20153787_

Round 1

Reviewer 1 Report

This is a well-written concise review providing a general overview about the LXRs role in polyunsaturated fatty acid metabolism. The manuscript will be better with incorporation of following minor comments. 1) Line 128 and 129. The sentence is confusing. Please consider revising. 2) Section 3: LXRs in phospholipid transfer and remodeling: The heading and subheading lacks brief background/context. The author directly jumped to the subheading on LPCAT3. 3) Figure 2 image quality need to be improved.

Author Response

Reviewer 1

This is a well-written concise review providing a general overview about the LXRs role in polyunsaturated fatty acid metabolism. The manuscript will be better with incorporation of following minor comments.

1) Line 128 and 129. The sentence is confusing. Please consider revising.

This has been revised : “PLs are continuously remodeled through deacylation and reacylation reactions. This process is called the Lands cycle”

2) Section 3: LXRs in phospholipid transfer and remodeling: The heading and subheading lacks brief background/context. The author directly jumped to the subheading on LPCAT3.

A brief background/context has been added (lines 123 to 129 of the revised manuscript)

 3) Figure 2 image quality need to be improved.

A higher resolution image will be provided a a separate file.

Reviewer 2 Report

The authors have set out to describe the role played by LXR in the regulation of PUFA synthesis and in phospholipid homeostasis. The manuscript is well written, clear and concise. The manuscript is timely as hepatic steatosis is increasing in prevalence.

Figure 2 Y axis should be scaled to show peak intensity. A scale bar should also be included to allow reader to determine how far away peaks are from the transcription start site.

Fig 2 should be referred to in text at line 136

Clarify in text if statement on 139 (re LPCAT not regulated by SREBP1c) is supported by USCS genome tracks in Fig 2. There is a small peak at the promoter, but without appropriate scaling of the Y axis, it is unclear if this a possible binding site.  

In fact, it appears as though SREBP1c is binding with similar peak intensity to both PLTP and LPCAT in the GW treated macrophages, as shown in Fig 2, yet in the text PLTP is described as a SREBP1c target and LPACT is not. This should be clarified in the text as Fig 2 doesn't support the argument in current form.

Line 147 - specify which synthetic agonist and dose

Line 167 - what concentration of T0 was used 

Line 197 -could the authors clarify what they mean by "Cis". it is unusual to capitlise this word, and typically a transcription factor would be trans-acting. The cited reference does not refer to simple transcription factor binding to a promoter and enacting gene expression (trans), but rather the LXR-DNA complex is acting in cis to regulate chromatin architecture downstream. This should be clarified in the text. 

There should be section/paragraph that explicitly describes which PUFA and PL are LXR ligands and how PUFA/PL regulation of LXR can feedback positively and negatively on PUFA/PL metabolism. The authors should comment on any in vivo evidence for how PUFA metabolism is regulated by endogenous (oxysterol) and dietary modulators of LXR.  

Minor points

Line 144 typo - 'amphipatic'

There is inconsistency in referencing, sometimes citations come after the period, sometimes before, sometimes with a space, sometimes without. This should be corrected. 

Gene name italics and lower/uppercase is also not clearly following standard rules: Ncor and NcoR line 93; Elovl5 and Elovl5. The manuscript should be carefully checked for similar errors.

Author Response

Reviewer 2

Comments and Suggestions for Authors

The authors have set out to describe the role played by LXR in the regulation of PUFA synthesis and in phospholipid homeostasis. The manuscript is well written, clear and concise. The manuscript is timely as hepatic steatosis is increasing in prevalence.

Figure 2 Y axis should be scaled to show peak intensity. A scale bar should also be included to allow reader to determine how far away peaks are from the transcription start site.

The figure has been modified. Y axis has been scaled and scale bar has been added for the X axis

Fig 2 should be referred to in text at line 136

Figure 2 has been referred line 136

Clarify in text if statement on 139 (re LPCAT not regulated by SREBP1c) is supported by USCS genome tracks in Fig 2. There is a small peak at the promoter, but without appropriate scaling of the Y axis, it is unclear if this a possible binding site.  In fact, it appears as though SREBP1c is binding with similar peak intensity to both PLTP and LPCAT in the GW treated macrophages, as shown in Fig 2, yet in the text PLTP is described as a SREBP1c target and LPACT is not. This should be clarified in the text as Fig 2 doesn't support the argument in current form.

This has been clarified :

Line 149 : LPCAT3 does not seem to be regulated by SREBP1c as SREBP1c knockdown does not affect LPCAT3 induction following LXR agonist treatment (ref 17)

Line 147 - specify which synthetic agonist and dose

This has been added  (T0901317, 10 mg/kg)

Line 167 - what concentration of T0 was used

This has been specified : 0.0075% T0901317 in the diet for 6 days

Line 197 -could the authors clarify what they mean by "Cis". it is unusual to capitlise this word, and typically a transcription factor would be trans-acting. The cited reference does not refer to simple transcription factor binding to a promoter and enacting gene expression (trans), but rather the LXR-DNA complex is acting in cis to regulate chromatin architecture downstream. This should be clarified in the text.

We agree the sentence was not clear, we changed to “ Finally, LXR suppresses inflammation through direct repression of pro-inflammatory genes”

There should be section/paragraph that explicitly describes which PUFA and PL are LXR ligands and how PUFA/PL regulation of LXR can feedback positively and negatively on PUFA/PL metabolism. The authors should comment on any in vivo evidence for how PUFA metabolism is regulated by endogenous (oxysterol) and dietary modulators of LXR. 

As suggested, we added a small paragraph regarding the regulation of LXR by PUFAs and its impact on fatty acid metabolism. Moreover, we discuss a study describing how dietary PUFAs may affect LXR activity and the potential consequences on hepatic steatosis (Line 203 to 207 of the revised manuscript)

Minor points

Line 144 typo - 'amphipatic'

This has been changed

There is inconsistency in referencing, sometimes citations come after the period, sometimes before, sometimes with a space, sometimes without. This should be corrected.

This has been corrected

Gene name italics and lower/uppercase is also not clearly following standard rules: Ncor and NcoR line 93; Elovl5 and Elovl5. The manuscript should be carefully checked for similar errors.

This has been checked. Italics have been used when it was explicitly referred to the gene. Lowercases have been used when it was explicitly referred to a mouse study.